# Effects of Dietary Sodium and Chloride on Slaughter Performance, Digestive Tract Development and Tibia Mineralization of Geese

**DOI:** 10.3390/ani13040751

**Published:** 2023-02-19

**Authors:** Yuanjing Chen, Zhiyue Wang, Haiming Yang

**Affiliations:** 1College of Animal Science and Technology, Yangzhou University, Yangzhou 225009, China; 2College of Veterinary Medicine, Yangzhou University, Yangzhou 225009, China

**Keywords:** sodium, chloride, goose, slaughter performance, digestive tract development, tibia quality

## Abstract

**Simple Summary:**

Sodium and chloride are essential electrolytes in the organism, mainly found in extracellular fluid, that play a critical role in maintaining osmotic pressure and acid-base balance of bodily fluids. However, excessive addition of sodium and chloride may cause many adverse effects, such as increasing water consumption and land salinization, and even threatening poultry health. This research explored the effects of different levels of sodium (0.10%, 0.15%, and 0.20%) and chloride (0.15%, 0.20%, and 0.25%) in diet on slaughter performance, intestinal development and tibia quality of 29–70-day-old geese. Our findings suggest a significant interaction between sodium and chloride on tibia quality; high sodium and chloride levels reduced the tibia’s strength. In addition, reducing dietary sodium and chloride levels will not have adverse effects on geese. The nutritional requirements for geese have yet to be determined. Therefore, the experiment provides new insights into the formulation of the diets of geese, especially the requirement of macronutrients.

**Abstract:**

This study evaluated the slaughter performance, digestive tract development and tibia mineralization effects of sodium (Na) and chloride (Cl) on geese. Four hundred and thirty-four male geese at 29 days were randomly assigned into nine groups with six replicates (eight in each). The experiment employed a 3 × 3 factorial design, with two instances each of three Na levels (0.10%, 0.15%, and 0.20%) and three Cl levels (0.15%, 0.20%, and 0.25%). All experimental birds were husbanded for 42 days. Dietary Na and Cl levels and their interactions (Na ×Cl) had no significant effect on the slaughter, breast, thigh, abdominal fat yield, and digestive tract index of geese (*p* > 0.05). However, dietary Na and Cl level significantly affected the crypt depth of the jejunum and tibial development. Variations in Na and Cl levels had a significant interaction on the crypt depth of jejunal (*p* < 0.05), 0.20% Na × 0.25% Cl had a minor crypt depth. Dietary variations in Na and Cl significantly affected the tibial strength, and there was a significant interaction between them (*p* < 0.05). When Na and Cl were at their maximum (0.20% Na and 0.25% Cl), the strength of the tibia was the lowest. In addition, a single factor (Na or Cl) had no effect (*p* > 0.05), but its interaction significantly affected the calcium (Ca) content of bone (*p* < 0.05). When the Na and Cl levels were 0.15% and 0.15%, respectively, the Ca content in bone was the highest. These results suggest that dietary Na and Cl had interactive effects on geese, especially in the development of the tibia. High dietary Na and Cl levels adversely influenced the tibia and intestinal crypt morphology. Therefore, we do not advocate supplementing too much Na or Cl in the diet. Combined with our previous results, for 29–70-day-old geese, it is recommended that dietary Na and Cl levels should be 0.10% and 0.15%, respectively.

## 1. Introduction

One of the most prevalent health problems in intensive poultry farming is tibial malformations and anomalies, which pose a serious hazard to the growth and development of poultry [1]. It can cause difficulty in poultry movement, affect feed intake, reduce meat quality, and cause significant economic losses to the poultry industry [2]. Dietary electrolytes like sodium (**Na**), potassium (**K**), and chloride (**Cl**) are essential because they keep the organism’s osmotic pressure and acid-base balance in check [3]. Dietary electrolyte balance (DEB) = Na + K − Cl was first established by Mogin [4] because it was believed that only these three elements had a significant role in acid-base balance. The lack of Na and Cl results in a significant decrease in osmotic pressure, which leads to water loss or dehydration and a sharp decrease in poultry performance [5,6]. However, excessive Na and Cl can interfere with the organism’s normal function and cause poisoning. The influence mechanism of Na and Cl on tibial performance is multifaceted. It may be that the acid-base balance affects the metabolic pathway of calcium, or acid-alkalosis leads to a nitrogen imbalance in animals, which causes the appearance of stubby tibias in broiler chickens, induces death syndrome and ascites syndrome, and the ability of animals to resist stress decreases [7,8].

In previous studies, dietary changes in Na and Cl did not influence the body weight of 29–70-day-old geese [9]. However, high dietary Na and Cl levels could increase water consumption and result in higher excreta moisture content. It is undoubtedly a negative effect, increasing water usage and potentially increasing the risk of many diseases [10], and the first effects may be leg and foot deformities. Wet litter enhanced the frequency and severity of footpad dermatitis [11]. Grazing or free-ranging geese may not be susceptible to litter moisture levels, however, indoor feeding methods may still lead to the occurrence of leg diseases. There were also studies showing that the shinbone of broilers with low excreta moisture content performed better [12,13]. Murakami et al. [14] showed that tibial mineralization increased with dietary Cl levels and was not affected by Na levels. However, Jankowski et al. [15] suggested that tibial mineralization decreased with improving Na levels but was not influenced by Cl levels. The gut, an essential organ responsible for nutrient absorption and waste excretion, is also affected by Na and Cl. Significant changes in intestinal absorption were observed in laying hens that were fed a diet low in salt, according to Elbrond et al. [16]. The number and surface area of epithelial cells increased after three to four weeks following a reduced sodium diet, which caused the micro-villi surface area to double. This led to an increase in Na channel density and ion exchange, which significantly improved net Na transport. Thus, the research aimed to expound the effects of dietary Na and Cl levels on the slaughter performance, intestinal morphology and tibial performance of geese. These results are expected to provide a reference for the rational use of NaCl in 29–70-day-old geese, and it is expected that the use of dietary sodium and chloride can be reduced without affecting the growth of geese, which will be beneficial to environmental protection.

## 2. Materials and Methods

### 2.1. Experimental Design, Diet, and Management

Protocols of the experiments were reviewed and approved by the Institutional Animal Care and Use Committee of Yangzhou University to ensure compliance with welfare and humane practices (SYXK [Su] IACUC 2018-0069).

All the geese in this study were fed at the Yangzhou University Experimental Farm. Geese were maintained under natural daylight conditions and kept at a room temperature of 20 ± 3 °C. Water and feed were provided ad libitum during the experiment. All Geese were reared in a plastic net bed (pen size: 2.10 m × 1.20 m, 2.52 m^2^).

Four hundred and thirty-four 29-day-old male Jiangnan White geese with uniform body weight (1.450 ± 0.13 kg) were obtained from Jiangsu Lihua Animal Husbandry Co., Ltd. (Changzhou, China). Their feeding program is the same until 29 days of age. The experimental design was completely randomized with a three × three factorial arrangement of treatments. Geese were randomly distributed into the 9 groups with 6 replicates (eight in each). G1, G2, G3 identical in Na content (0.10%); G4, G5, G6 identical in Na content (0.15%); and G7, G8, G9 identical in Na content (0.20%). G1, G4, G7 identical in Cl content (0.15%); G2, G5, G8 identical in Cl content (0.20%); and G3, G6, G9 identical in Cl content (0.25%). The experiment lasted for 42 days. The DEB of each group was kept at 251 mEq/kg. The experimental diets (Table 1) were prepared on the basis of the National Research Council (NRC) for geese older than four weeks [17] and existing studies from our department [18]. Before the experiment, the concentrations of crude protein (CP) and crude fiber (CF) in each feedstuff were analyzed in triplicate according to the guidelines of the Association of Official Analytical Chemists (AOAC, 2000) [19]. Na and K were analyzed by flame photometry, while Cl was analyzed by titration with AgNO_3_ [20]. Calcium (Ca) was determined by atomic absorption spectrophotometry, Phosphorus (P) was determined spectrophotometrically from a filtered ash solution using the vanadium molybdate method [19]. We regulated their diet’s Na and Cl levels by supplementing NaCl, K_2_SO_4_, NaHCO_3_, KCl and NH_4_Cl.

### 2.2. Sample Collection

#### 2.2.1. Carcass Measurements

At 70 days, one goose per replicate (54 geese in total) was selected randomly for slaughter by exsanguination. The slaughter weight (after exsanguination and defeathering), eviscerated carcass weight, abdominal fat, breast and leg were separated and recorded. In relation to body weight, the slaughter yield and eviscerated carcass yield were computed. The abdominal fat, breast, and leg yield were calculated relative to the eviscerated carcass weight.

#### 2.2.2. Digestive Tract Index

Six geese from each treatment (54 geese in total) were selected on day 70, slaughtered, and intestinal samples were collected. After trimming the fat and chyme, the absolute weights of the proventriculus, gizzard, duodenal, jejunum, ileum (Meckel’s diverticulum was taken as the dividing point between the jejunum and ileum) and cecum were recorded and expressed as a percentage of body weight (BW).

#### 2.2.3. Intestinal Morphology

One goose per replicate (54 geese in total) was used for intestinal morphological examination. About 3 cm long sections were cut from the midpoint of the duodenum, jejunum, and ileum, immediately washed with ice-cold saline and immersed in 10% neutral buffered formalin for fixation. Samples were dehydrated through ascending alcohol concentrations (60%, 75%, 85%, 95%, and 100%) and embedded in paraffin, cut into 5 μm thick sections with a Microtome (Leica, Weztlar, Germany), stained with hematoxylin-eosin, and examined by a Zeiss Axiophot microscope (Carl Zeiss, Ober kochen, Germany). Villus height and crypt depth were measured visually on ten villi using a computer-assisted image analysis program (LY-HPCCD5, Liyang Ltd., Chengdu, China) at 40× magnification. The variables measured were villus height (VH), crypt depth (CD), and a calculation of the VH:CD (villus height/crypt depth) ratio.

#### 2.2.4. Bone Strength: Breaking Bones 

After the geese were slaughtered (one goose per replicate, 54 geese in total), tibias were collected from the right side (standing state) of the body. The weight, length, width and thickness of the tibias were measured. The meat and fascia were removed before bone-breaking strength analyses. Tibia bone-breaking strength was determined with a double-column desktop electronic testing machine (In-stron3367, Instron Corporation, Norwood, MA, USA). The deformation rate was 5 mm/min. The tibia was placed on the bracket at the same bending degree, and the upper pressure was loaded at a uniform speed until the tibia was broken and the bending load (Newton, N) at the time of fracture was recorded.

#### 2.2.5. Bone Mineral Content

After removing the attachment on the right tibia, bone mineral content was measured. The tibia was dried at 105 °C for 48 h then defatted with ether for seven days and dried to constant weight at 105 °C. The dried bones were crushed and put in the crucible then carbonized on an electric furnace until smokeless and burned in a muffle furnace at 550–600 °C for 24 h (RC-MF12-30, Beijing Ruicheng Yongchuang Technology Co., Ltd., Beijing, China). After obtaining the tibia ash, it was made into sample decomposition liquid. The calcium (Ca) was analyzed by the Ethylene Diamine Tetraacetie Acid (EDTA) method according to the GB/T6436–2018. The phosphorus (P) was determined via the molybdate-yellow colorimetry method based on the GB/T6437–2018.

### 2.3. Statistical Analysis

All the statistical evaluations were performed in the software package SPSS 21.0 (SPSS Inc., Chicago, IL, USA). The data were analyzed as a three × three (Na × Cl) factorial arrangement of treatments by a two-way ANOVA with a model that included the main effects of Na and Cl and their interaction. Linear and quadratic effects were also analyzed. The means were compared by Duncan’s multiple comparison tests to determine whether there were significant effects between them. *p* < 0.05 was used as the criterion for statistical significance.

## 3. Results

### 3.1. Slaughter Performance and Digestive Tract Index

Na, Cl and Na × Cl had no effect on the yield of abdominal fat, breast, thigh, or the eviscerated carcass (*p* > 0.05). Dietary Cl levels tended to affect the slaughter yield of geese (*p* = 0.052) (Table 2). Dietary Na levels had no significant effect on the proventriculus index of geese (Table 3; *p* > 0.05), but there was an increasing trend with dietary Cl level (*p* = 0.063). Varying dietary Na and Cl levels had no significant influence on gizzard, duodenal, jejunum, ileum or cecum index (*p* > 0.05). No significant interactions (*p* > 0.05) between dietary Na and Cl supplementation were observed on the gizzard, duodenal, jejunum, ileum and cecum index of geese.

### 3.2. Intestinal Morphology

The effect of dietary Na and Cl on morphological parameters of the duodenum, jejunum and ileum was given in Table 4. With the increase of dietary Na, there was an increasing trend in the wall thickness of duodenal (*p* = 0.070), but Cl level and Na × Cl had no effects on it (*p* > 0.05). There was a trend in the wall thickness of jejunum being influenced by Na × Cl (*p* = 0.073) rather than Na and Cl levels alone (*p* > 0.05). Dietary Na and Cl levels had significant effects on the crypt depth of the jejunum (*p* < 0.05), with the 0.20% Na × 0.25% Cl (G9) having the shortest crypt depth of 261.33 μm. However, there was no significant difference between 0.20% Na × 0.25% Cl (G1), 0.20% Na × 0.25% Cl (G6), 0.20% Na × 0.25% Cl (G8) and 0.20% Na × 0.25% Cl (G9) (*p* > 0.05). With the increased dietary Na, the crypt depth of jejunum tended to decrease (*p* = 0.067). Dietary Na and Cl levels had no significant effects on villus length, crypt depth, VH:CD of duodenal, villus length, VH:CD of jejunum, wall thickness, villus length, crypt depth or VH:CD of ileum (*p* > 0.05).

### 3.3. Tibial Quality

Dietary Na and Cl levels and their interaction had significant effects on the breaking strength of the tibias of 70-day-old geese (*p* < 0.05) (Table 5). With increased Na levels, the thickness of the tibias initially decreased and then increased (*p* = 0.097). The breaking strength of tibias decreased significantly with the increased Na and Cl levels (*p* < 0.05). The breaking strength of tibias at the 0.10% Na level was 5.28% higher than at the 0.20% Na level. The breaking strength of tibias at the 0.15% Cl level was 1.91% higher than at the 0.25% Cl level. There was a significant interaction (*p* < 0.05) between dietary Na and Cl supplementation on the breaking strength of tibias, showing 0.20% Na × 0.25% Cl (G9) to be significantly lower than all groups except for 0.15% Na × 0.25% Cl (G6) (*p* < 0.05). The breaking strength of the tibias in the G3 group was significantly higher than that of other groups (*p* < 0.05).

The interaction of Na and Cl significantly affected the calcium content of tibias in geese (*p* < 0.05). When the dietary Na level was 0.15% and the Cl level was 0.15% (G4), the calcium content of tibias was the highest, 23.24%. The calcium content of the tibias displayed a declining trend when Cl levels rose (*p* = 0.053) and with the increased of Na levels, it initially increased and then decreased (*p* = 0.070). With the addition of Cl, The phosphorus content of the tibias tended to increase first and then decrease (*p* = 0.076). Dietary Na, Cl and Na × Cl had no significant effects on weight, length and width of tibia in geese (*p* > 0.05).

## 4. Discussion

### 4.1. Slaughter Performance and Digestive Tract Index

In this experiment, the eviscerated carcass output of 70-day-old geese was unaffected by Na or Cl levels. However, the slaughter yield tended to increase with the dietary Cl level. Similarly, Mushtaq et al. [21] reported that a quadratic effect of dietary Cl was found in the slaughter yield of broilers (*p* < 0.001). The slaughter yields of 0.30%, 0.40%, and 0.50% Cl groups were 60.91%, 63.60% and 59.43%, respectively. The results may be due to the high water content in the carcass. Teeter et al. [22] confirmed that when increased electrolyte in the broiler diet, the water absorption of the carcass increases. Mushtaq et al. [23] reported that dietary Cl had no effect on slaughter yield, breast yield, thigh yield, and abdominal fat yield of broilers. Yasoob et al. [24] showed that different dietary sodium bi-carbonate levels had no significant effect on slaughter, thigh and abdominal fat yield of broilers at 42 d. Bidar et al. [25] showed that dietary Na intake had no effect on the slaughter performance of broilers, or its effect was limited to reducing abdominal fat deposition [26]. Our results also found that Na and Cl levels did not affect the breast yield, thigh yield, and abdominal fat yield. Overall, this result confirmed that low Na and Cl levels had no adverse effects on slaughter performance.

### 4.2. Digestive Tract Index

Dietary variations in Na and Cl had no effect on the proventriculus, gizzard and intestinal index of geese, similar to the results of Mushtaq et al. [23]. This result may be because the feed intake did not differ between the groups. Jankowski [15] found that the gizzard index of broilers at 35 d in the NaCl-free group was significantly lower than that of the NaCl-supplemented group. The relative weight of the small intestine in the group without NaCl addition was significantly lower than that of the group with the maximum NaCl addition. Furthermore, the feed intake of the NaCl-free group was significantly lower than that of the NaCl-supplemented group. The feed intakes of the NaCl-free group in 15–35 days were 0.127%, 0.254%, 0.382%, 0.509%, and 0.636%; whereas for the NaCl-supplemented group they were 1299.48 g, 2763.96 g, 2838.93 g, 2829.4 g, 2910.24 g and 2968.44 g, respectively. However, the gizzard index and feed intake did not differ between the NaCl-supplemented groups (0.127–0.636% NaCl). The structure and function of the intestine gradually matures with increasing animal age and change of feed intake [27]. When the volume of feed consumed exceeds the volume capacity of the digestive tract, the digestive organs gradually change to accommodate large volumes of dietary feed intake [28]. When there is no significant change in feed intake, the digestive tract index is difficult to change.

### 4.3. Intestinal Morphology

Nutrient digestion and absorption occur mainly in the small intestine, thus intestinal health directly affects the poultry digestion, absorption and metabolism of nutrients [29]. In particular, villus height, crypt depth and wall thickness of the small intestine are essential indicators for measuring the digestion and absorption function [30]. The influence of Na and Cl on intestinal morphology parameters has been scarcely investigated. In our study, dietary Na and Cl levels only had a significant interaction effect on the crypt depth of the jejunum; and when the Na and Cl levels were the highest (G9), the crypt depth of jejunum was the smallest. VH:CD did not differ between the groups, although Na and Cl showed a tendency to affect some indicators (the wall thickness of duodenal and jejunum). It may also be because there was no difference in feed intake among the groups, as Li et al. [31] reported that broilers in the Feed groups had a higher villus height of the mid small intestine than the Fast groups at days 2 and 7. Changes in crypt depth morphologically characterize the growth rate of crypt cells. The recess depth represents the pace of cell division, and crypt shallowness denotes a faster rate of intestinal epithelial cell maturation and improved absorption capacity [32]. Deeper crypts mean a faster proliferation of crypt cells. The electrolyte change may affect the replacement of intestinal cells, but this conclusion needs further verification.

Broilers in the Feed groups had a higher villus height of the mid small intestine than the Fast groups at days 2 and 7.

### 4.4. Tibial Quality

During the growth and development of poultry, the order of tissue development is bone then muscle and fat. Among them, bone, the preferential developmental tissue of poultry bodies, plays a critical role in the growth process [33]. The development of the tibia is of great significance to the growth of poultry and has a significant influence on their performance potential [34,35]. As the main components of poultry bones, calcium (Ca) and phosphorus (P) significantly impact the bones’ normal development. Too high or too low levels will cause bone metabolic diseases [36]. Jankowski et al. [15] found that dietary Na and Cl levels had no significant effect on the tibial weight of the broiler, and the tibial strength of the treatment group without Na and Cl supplementation was the lowest. Nevertheless, in the treatment group supplemented with Na and Cl, the strength of tibia decreased linearly with Na and Cl. When Na and Cl were at the maximum level (0.261% Na and 0.481% Cl), the strength of the tibia was the lowest; the tibial strength was greatest when Na and Cl were at the minimum level (0.074% Na and 0.198% Cl). Sayed et al. [37] showed that improving Na levels from 0.20% to 0.30% significantly decreased tibial weight and the calcium and phosphorus content of the bone. In agreement with the current study, our results showed that both Na and Cl levels effect tibial strength and there is a very significant interaction. With raised Na and Cl levels the breaking strength (N) of tibias decreased significantly. This proves that there is a synergistic effect between Na and Cl manifesting in the Ca content of the bone. A single factor (Na or Cl) tended to affect the Ca levels in the bone but did not differ significantly. However, the interaction of the Na and Cl levels significantly affected the Ca level in the bone. When the Na and Cl levels were both 0.15%, the calcium content of the bone was the highest. The degree of physiological reaction to toxic or inadequate levels might vary due to the numerous metabolic and absorptive interactions among the macronutrients. Owing to these interactions, it is challenging to calculate the optimal dietary intake of each macronutrient needed for poultry. The mechanisms of Na and Cl effects on tibial development are multifaceted, possibly due to DEB affecting Ca metabolism or excessive Cl causing acidosis. Leach et al. [38] showed that a diet with a high Cl level increased the incidence and severity of tibial osteochondrosis in broilers. With the deepening of research, increased evidence shows that a high electrolyte intake can cause bone loss and lead to osteoporosis and fractures. The mechanism is mainly due to the increased Ca excretion caused by the intake of electrolytes, which causes a Ca imbalance in the organism and leads to bone loss [39]. These results also suggest that a simultaneous increase in dietary Na and Cl may have a greater negative impact on the tibia of geese.

## 5. Conclusions

In conclusion, Na and Cl had interactive effects on the crypt depth of jejunum, as well as the strength and calcium content of tibias in geese. High Na or high Cl decreased the crypt depth of jejunum and the strength and calcium content of tibias. This suggests that too much Na and Cl in the diet could negatively affect geese, especially if they are high in both. Combined with our previous results, for 29–70-day-old geese, it is recommended that dietary Na and Cl levels be 0.10% and 0.15%, respectively.

## Figures and Tables

**Table 1 animals-13-00751-t001:** Composition and nutrient level of experimental diet in geese (air-dry basal).

Item	Treatments								
	G1	G2	G3	G4	G5	G6	G7	G8	G9
Ingredient (%)									
Corn	59.96	59.94	59.70	59.97	59.93	59.90	59.91	59.95	59.75
Soybean meal	26.36	26.33	26.40	26.34	26.33	26.37	26.35	26.33	26.37
Limestone	1.20	1.20	1.20	1.20	1.20	1.20	1.20	1.20	1.20
Rice husk	9.09	9.05	9.06	9.09	9.05	9.03	9.05	9.07	9.03
NaHCO_3_	0.37	0.11	0.36	0.39	0.27	0.29	0.75	0.52	0.59
NaCl	0.00	0.18	0.00	0.12	0.20	0.18	0.00	0.16	0.11
KCl	0.32	0.22	0.36	0.17	0.23	0.30	0.19	0.24	0.16
K_2_SO_4_	0.39	0.66	0.59	0.37	0.49	0.47	0.16	0.23	0.40
NH_4_Cl	0.00	0.00	0.11	0.00	0.00	0.00	0.10	0.00	0.14
CaHPO_4_	1.15	1.15	1.15	1.15	1.15	1.15	1.15	1.15	1.15
DL-Methionine	0.16	0.16	0.16	0.16	0.16	0.16	0.16	0.16	0.16
Premix^1^	1.00	1.00	1.00	1.00	1.00	1.00	1.00	1.00	1.00
Total	100.00	100.00	100.00	100.00	100.00	100.00	100.00	100.00	100.00
Nutrient levels									
Metabolic energy (MJ/kg)	10.81	10.81	10.78	10.81	10.81	10.80	10.80	10.81	10.78
Crude protein (%)	17.01	17.00	17.01	17.00	17.00	17.01	17.00	17.00	17.00
Crude fiber (%)	6.53	6.51	6.51	6.52	6.51	6.50	6.51	6.52	6.50
Na (%)	0.10	0.10	0.10	0.15	0.15	0.15	0.20	0.20	0.20
Cl (%)	0.15	0.20	0.25	0.15	0.20	0.25	0.15	0.20	0.25
K (%)	0.97	1.04	1.08	0.89	0.97	1.00	0.80	0.86	0.89
DEB (mEq/kg)	251.80	251.80	252.10	251.60	251.90	251.80	251.90	251.80	251.70
Ca (%)	0.91	0.91	0.91	0.91	0.91	0.91	0.91	0.91	0.91
Total P (%)	0.60	0.60	0.60	0.60	0.60	0.60	0.60	0.60	0.60
Lysine (%)	0.84	0.84	0.84	0.84	0.84	0.84	0.84	0.84	0.84
Methionine (%)	0.41	0.41	0.41	0.41	0.41	0.41	0.41	0.41	0.41

^1^ Vitamins and minerals premix used to cover the required vitamins and minerals per kilogram of diet (VA, 12,000 IU; VD_3_, 4000 IU; VE, 18 IU; VK, 0.15 g; VB_1_, 0.06 g; VB_2_, 0.6 g; VB_6_, 0.2 g; VB_12_, 1 mg; nicotinic acid, 3 g; D-pantothenic acid, 0.9 g; folic acid, 0.05 g; biotin, 4 mg; Zn, 9 g; Mn, 9.5 g; Cu, 1 g; Se, 30 mg; I, 50 mg; Fe, 6 g).

**Table 2 animals-13-00751-t002:** Effects of dietary Na and Cl on slaughter performance (%) of 70-day-old geese (*n* = 6).

Item	Dietary Na, %	Dietary Cl, %	Slaughter Yield ^1^	Eviscerated Carcass Yield ^1^	Breast Yield ^2^	Thigh Yield ^2^	Abdominal Fat Yield ^2^
G1	0.10	0.15	89.02	72.53	11.04	15.07	3.66
G2	0.10	0.20	88.75	71.30	10.85	14.43	3.82
G3	0.10	0.25	90.10	73.42	11.26	15.70	3.22
G4	0.15	0.15	87.88	71.76	10.95	15.70	3.23
G5	0.15	0.20	88.85	71.09	11.18	15.55	4.04
G6	0.15	0.25	88.88	70.91	11.79	14.54	3.53
G7	0.20	0.15	88.30	72.09	11.10	15.56	3.03
G8	0.20	0.20	88.27	72.12	11.18	15.35	3.63
G9	0.20	0.25	89.40	72.41	11.57	15.65	3.23
Na	0.10		89.29	72.42	11.05	15.07	3.57
	0.15		88.54	71.25	11.31	15.27	3.60
	0.20		88.66	72.21	11.28	15.52	3.30
Cl		0.15	88.40	72.13	11.03	15.44	3.31
		0.20	88.62	71.50	11.07	15.11	3.83
		0.25	89.46	72.25	11.54	15.30	3.32
SEM			0.188	0.239	0.135	0.174	0.122
*p*-value	Na		0.201	0.109	0.707	0.578	0.549
	Linear		0.170	0.715	0.485	0.298	0.372
	Quadratic		0.272	0.038	0.625	0.941	0.518
	Cl		0.052	0.387	0.270	0.743	0.156
	Linear		0.020	0.836	0.128	0.731	0.955
	Quadratic		0.427	0.184	0.450	0.494	0.047
	Na × Cl		0.717	0.475	0.963	0.277	0.845

^1^ Calculated as a percentage of live body weight. ^2^ Calculated as a percentage of eviscerated carcass weight.

**Table 3 animals-13-00751-t003:** Effects of dietary Na and Cl on digestive tract index of 70-day-old geese (*n* = 6).

Item	Na, %	Cl, %	Proventriculus ^1^	Gizzard ^1^	Duodenal ^1^	Jejunum ^1^	Ileum ^1^	Cecum ^1^
G1	0.10	0.15	2.72	29.10	2.57	5.30	5.23	1.45
G2	0.10	0.20	2.78	28.86	2.63	5.00	4.43	1.69
G3	0.10	0.25	3.10	30.09	2.36	4.84	4.71	1.40
G4	0.15	0.15	2.53	30.05	2.61	4.76	4.83	1.44
G5	0.15	0.20	2.75	30.99	2.82	5.93	6.07	1.39
G6	0.15	0.25	2.93	30.77	2.35	5.87	4.64	1.56
G7	0.20	0.15	2.64	29.54	2.55	5.14	4.64	1.63
G8	0.20	0.20	3.09	31.82	2.64	5.48	5.49	1.54
G9	0.20	0.25	2.62	31.19	2.73	5.64	5.14	1.64
	0.10		2.87	29.35	2.52	5.05	4.79	1.51
Na	0.15		2.74	30.60	2.59	5.52	5.18	1.46
	0.20		2.78	30.85	2.64	5.42	5.09	1.60
		0.15	2.63	29.56	2.57	5.07	4.90	1.51
Cl		0.20	2.87	30.56	2.70	5.47	5.33	1.54
		0.25	2.89	30.68	2.48	5.45	4.83	1.53
SEM			0.052	0.451	0.039	0.158	0.148	0.037
*p*-value	Na		0.559	0.384	0.384	0.406	0.452	0.504
	Linear		0.512	0.181	0.181	0.217	0.337	0.413
	Quadratic		0.450	0.603	0.603	0.903	0.401	0.442
	Cl		0.063	0.570	0.570	0.053	0.517	0.311
	Linear		0.042	0.320	0.320	0.304	0.327	0.854
	Quadratic		0.288	0.654	0.654	0.034	0.539	0.143
	Na × Cl		0.100	0.919	0.919	0.100	0.488	0.112

^1^ Calculated as a percentage of live body weight.

**Table 4 animals-13-00751-t004:** Effects of dietary Na and Cl on the intestinal morphology of 70-day-old geese (*n* = 6).

Item	Na, %	Cl, %	Duodenal				Jejunum				Ileum			
			Wall Thickness	Villus Height	Crypt Depth	VH:CD	Wall Thickness	Villus Height	Crypt Depth	VH:CD	Wall Thickness	Villus Height	Crypt Depth	VH: CD
G1	0.10	0.15	263.35	615.17	280.49	2.20	316.77	668.36	266.07 ^ab^	2.52	315.61	628.51	269.51	2.37
G2	0.10	0.20	271.35	613.47	274.62	2.24	328.54	671.83	276.10 ^c^	2.44	317.73	668.05	251.88	2.66
G3	0.10	0.25	263.51	609.03	267.63	2.28	304.52	689.86	279.02 ^c^	2.48	315.71	667.66	265.61	2.58
G4	0.15	0.15	273.95	623.96	265.19	2.36	319.15	672.88	274.06 ^bc^	2.46	309.49	614.19	259.53	2.39
G5	0.15	0.20	266.72	623.59	279.32	2.25	311.41	661.53	274.09 ^bc^	2.41	303.80	676.65	268.06	2.53
G6	0.15	0.25	277.19	611.97	287.02	2.17	308.23	644.54	265.34 ^ab^	2.43	308.40	637.39	248.16	2.62
G7	0.20	0.15	271.30	612.35	272.85	2.26	289.57	648.46	271.98 ^bc^	2.38	304.59	677.03	263.44	2.59
G8	0.20	0.20	275.76	624.12	275.37	2.28	310.33	661.18	269.81 ^abc^	2.45	299.05	677.47	249.20	2.74
G9	0.20	0.25	279.89	613.29	275.52	2.25	327.06	655.12	261.33 ^a^	2.51	316.61	686.21	239.49	2.86
Na	0.10		266.07	612.56	274.25	2.24	316.61	676.68	273.73	2.48	316.35	654.74	262.33	2.53
	0.15		272.62	619.84	277.17	2.26	312.93	659.65	271.16	2.43	307.23	642.74	258.58	2.51
	0.20		275.65	616.59	274.58	2.26	308.99	654.92	267.71	2.45	306.75	680.24	250.71	2.73
Cl		0.15	269.53	617.16	272.84	2.27	308.49	663.23	270.70	2.45	309.90	639.91	264.16	2.45
		0.20	271.27	620.39	276.43	2.26	316.76	664.85	273.33	2.43	306.86	674.06	256.38	2.65
		0.25	273.53	611.43	276.72	2.23	313.27	663.17	268.56	2.47	313.57	663.75	251.08	2.69
SEM			1.728	2.987	3.062	0.027	3.486	4.439	1.204	0.019	2.769	7.778	3.208	0.054
*p*-value	Na		0.070	0.639	0.919	0.936	0.659	0.117	0.067	0.682	0.311	0.134	0.318	0.205
	Linear		0.023	0.588	0.966	0.717	0.382	0.045	0.042	0.717	0.161	0.178	0.144	0.137
	Quadratic		0.622	0.414	0.678	0.931	0.986	0.506	0.860	0.931	0.463	0.132	0.763	0.296
	Cl		0.626	0.503	0.859	0.823	0.610	0.985	0.178	0.739	0.633	0.184	0.246	0.167
	Linear		0.355	0.439	0.613	0.522	0.582	0.996	0.469	0.698	0.594	0.210	0.100	0.071
	Quadratic		0.945	0.343	0.803	0.957	0.435	0.865	0.151	0.488	0.415	0.178	0.855	0.485
	Na × Cl		0.368	0.930	0.513	0.619	0.073	0.381	0.004	0.614	0.814	0.671	0.360	0.972

^a–c^ Values within a row with different superscripts differ significantly at *p* < 0.05. VH:CD, villus height/crypt depth.

**Table 5 animals-13-00751-t005:** Effects of dietary Na and Cl on tibial quality of 70-day-old geese (*n* = 6).

Item	Na, %	Cl, %	Tibia Weight (g)	Tibia Length (cm)	Tibia Width (cm)	Tibia Thickness (cm)	Breaking Strength (N)	Ca (%)	P (%)
G1	0.10	0.15	18.57	15.47	0.93	0.71	256.7^cd^	22.95^bc^	10.13
G2	0.10	0.20	17.50	15.26	0.93	0.69	253.0^bcd^	22.59^abc^	10.94
G3	0.10	0.25	19.17	15.63	0.92	0.68	268.6^e^	21.99^a^	10.67
G4	0.15	0.15	17.75	15.35	0.91	0.67	259.6^d^	23.24^c^	10.46
G5	0.15	0.20	17.93	15.53	0.95	0.67	247.3^b^	22.25^ab^	11.27
G6	0.15	0.25	17.33	15.09	0.91	0.67	245.1^ab^	22.36^ab^	10.13
G7	0.20	0.15	18.97	15.30	0.97	0.70	250.7^bc^	21.76^a^	10.01
G8	0.20	0.20	17.80	15.22	0.93	0.68	248.0^bc^	22.61^abc^	10.55
G9	0.20	0.25	17.65	15.43	0.93	0.66	238.3^a^	21.98^a^	10.66
Na	0.10		18.41	15.45	0.92	0.69	259.4^c^	22.51	10.58
	0.15		17.67	15.32	0.92	0.67	250.6^b^	22.62	10.62
	0.20		18.14	15.31	0.94	0.68	245.7^a^	22.12	10.41
Cl		0.15	18.43	15.37	0.93	0.69	255.6^b^	22.65	10.20
		0.20	17.74	15.33	0.93	0.68	249.4^a^	22.48	10.92
		0.25	18.05	15.38	0.92	0.67	250.7^a^	22.11	10.49
SEM			0.199	0.054	0.006	0.004	1.45	0.104	0.128
*p*-value	Na		0.303	0.507	0.369	0.097	<0.001	0.070	0.763
	Linear		0.577	0.307	0.233	0.165	<0.001	0.121	0.588
	Quadratic		0.158	0.596	0.461	0.100	0.467	0.167	0.642
	Cl		0.366	0.928	0.550	0.188	0.026	0.053	0.076
	Linear		0.442	0.935	0.343	0.081	0.160	0.034	0.351
	Quadratic		0.247	0.713	0.614	0.725	0.222	0.635	0.033
	Na × Cl		0.241	0.154	0.254	0.445	<0.001	0.017	0.510

^a–d^ Values within a row with different superscripts differ significantly at *p* < 0.05.

## Data Availability

Data are available upon reasonable request from the corresponding author.

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
