# Peer review of "Effects of Dietary Sodium and Chloride on Slaughter Performance, Digestive Tract Development and Tibia Mineralization of Geese"

_animals, 2023, doi:10.3390/ani13040751_

Round 1
Reviewer 1 Report
The paper entitled "Effects of Dietary Sodium and Chloride on Slaughter Performance, Digestive Tract Development and Tibia Mineralization of Geese" is according to the Aims & Scope of the Journal. The novelty can be partially discussed however, it is not without significance that the experiment was carried out on geese, in a topic that will enrich the existing literature. The experiment was well designed and described, however, it needs some important clarifications. As it was mentioned, the paper is interesting due to the choice of poultry species, it can enrich the current knowledge, although the generally presented ones are already partly known and confirmed. Estimated 80% of references are older than 5 years, however, it cannot be denied that in terms of the aforementioned research it is difficult to find other references corresponding to the subject, in particular that they concern geese. Considering this, I believe that the work can be published in "Animals", but requires a general revision and supplementation of key information. Please consider the comments below.
Simple Summary:
14 – Please add information that the experiment assumes different levels of Na and Cl.
17 – „calcium content” Note that the 0.20 Na and 0.25 Cl levels were no different from the 0.10 and 0.20 levels for example.
19 – Please change „configuration” to „formulation”
20 – Maybe "macronutrients" could be better. Mineral elements is quite a broad concept
Abstract:
23 – In what context did you use the word "healthy"? Please explain. Generally, such terms are not used, you omit them later in Materials and Methods
24-25 – Please indicate in any way about the combinations of Na and Cl in these mentioned diets. It gives the impression that there were 6 diets when there were actually 9, please consider re-description to clarify this point.
27 – No comma between "yield" and "and". Check out the entire manuscript in this regard.
28 – This sentence "affected intestinal morphology" - it's too general. The effect was only for crypt depth in jejunum, please verify it.
29-30 – Please verify, other levels of Na and Cl did not differ from those mentioned, e.g. 0.10 and 0.15
31-32 – similarly to 0.15 and 0.25, they did not differ from those mentioned
33-34 – G1, G2, G8 did not differ significantly from G4
36 – "intestine" too general
36-37 – please rephrase "Therefore, we do not advocate excessive supplementation of Na or Cl in the diet, especially too many."
39 – „be” to „should be” - consider a change
Introduction:
49 – Please change "body" to "organism" throughout the manuscript
59-60 – no reference to “In previous studies, dietary changes in Na and Cl had not influenced the body weight 59 of 29-70-day-old geese.”
63- Please specify "and the first effects may be leg and foot deformities." It seems to be about FPD and this problem in geese is not that important, or please specify it referring to the country of maintenance etc.. Differences can be really significant between regions, countries, this is important.
63-65 – In general, consider whether this passage is relevant, especially since the work is about geese, not broilers as above in general. You need to make it more specific, as per the previous comment.
71-73 „After three to four weeks of consuming a low sodium diet, the micro-villi surface area more than doubled as a result of an increase in the number and surface area of epithelial cells.” Relatively vague sentence, please rephrase
74 – expected effects, hypothesis?? This is a really important point!
Materials and Methods:
There are some important information gaps, in particular in section 2.1 (the subtitle should include diets). What were the geese?? Breed, crossbreed, where did they come from, how were they fed until day 29?? It is necessary to add information on how the birds were kept. Was a lighting program used? Where were they kept? Maintenance is extremely important! Did the birds have access to the enclosure? What was the bird density?
Also, the description of the diets is a bit confusing. Perhaps it is worth presenting in what configurations the diets were used (G1-G9). Which diets were identical in Na content? eg G1, G2, G3 had 0.10 etc... and G1, G1, G4, G5 identical in Cl content. Just to make it clear to the reader. Was there a control diet?
89-90 and 92 there is information in the text on chemical analyzes of diets, but it concerns Na, K, Cl. In table 1 there is information on "nutrients levels", for example, Crude protein, Crude fiber ect. But nowhere in the work does it show how these levels were determined. Analytical, calculations? There are also no results regarding the K analysis, and such were done.
111 – six geese from each treatment, i.e. in total, the average was from how many birds? In the table it is n=6 but this indicates that there were a total of 6 birds from each treatment one of the replications or how 6x6 so n=36? Please elaborate on this in each paragraph.
116 – how was Wall Thickness measured???
127 – where are the results regarding VH:CD ratio??
129 – from how many birds were bones taken, n=???
147 - how were the outliers removed, what were/was the criteria used?
Results:
156 – remove the subtitle "Growth performance" or add other titles as appropriate to the paragraphs
160-162 – there was a trend for Cl, please take this into account instead of using the word increased, which is assumed to imply a significant difference
162-163 please add where the interaction has not been noted.
167 – the tendency. Please mark it
169 – indicate which groups were not significantly different from 0.20 Na and 0.25 Cl
175- Please change "tibial strength" to breaking strength of tibia
176 – if there is a Na x Cl interaction, please report these results, not the Na and Cl main results
179-180 – I suggest, add in parentheses what group it was, it will be easier to locate it in the table. If so, apply to the entire manuscript.
181-182 – note that as the Na level increased, the calcium initially increased and then decreased. Verify this sentence again.
182-183 – verify this sentence, because according to table 5 there is a tendency (tibbia thickness for Na and P % for Cl)
213 – explanation of the abbreviations Ca and P seems unnecessary
Discussion:
217- it is worth adding a comparison to the level of 0.40
221-222 What a „previous observations”??
224 – What a„diffrent electorlyte??”
231 – it would be advantageous to provide information on feed intake between the groups. Feed consumption is an important parameter in this study and should be considered seriously.
233-234 – this sentence is unclear and needs re-verification „However, there was no difference among the levels in the NaCl supplemented group (0.127 % -0.509 %)”
249-250 - indicate that the result was significant only for jejunum
250-254 – an important parameter is the VH:CD ratio, I suggest adding it and discussing it as this parameter best characterizes the development of the intestines. You should definitely consider that.
274 – but there was a tendency for Na and Cl
Conclusions:
287- verify this for Ca concentration. G9 and, for example, G2 did not differ significantly from each other, and the level of Na was lower.
287-289 – too bold statement when only differences in crypt depth dudeanum and no ratio VH:CD were observed
290-291 – here you contradicted the previous sentence
291 – I would change "to sum up" to "in conclusion".
Author Response
Response to Reviewer 1 Comments
Thank you for your comments concerning our manuscript entitled “Effects of Dietary Sodium and Chloride on Slaughter Performance, Digestive Tract Development and Tibia Mineralization of Geese” (ID: animals-2155543). Those comments are all valuable and very helpful for revising and improving our paper, as well as the important guiding significance to our researches. We have studied comments carefully and have made correction which we hope meet with approval. Revised portion are marked in yellow in the paper. The main corrections in the paper and the responds to the comments are as flowing:
Simple Summary:
14 – Please add information that the experiment assumes different levels of Na and Cl.
Response: According to the reviewer’s comment, we have corrected the sentences “This research explored the effects of different levels of sodium (0.10%, 0.15%, and 0.20%) and chloride (0.15%, 0.20%, and 0.25%) in diet on slaughter performance, intestinal development and tibia quality of 29-70 days old geese.” in line 14-16.
17 “calcium content” Note that the 0.20 Na and 0.25 Cl levels were no different from the 0.10 and 0.20 levels for example.
Response: The correction has been made.
19 – Please change “configuration” to “formulation”
Response: The correction has been made.
20 – Maybe "macronutrients" could be better. Mineral elements is quite a broad concept
Response: Thank you for your valuable and thoughtful comments. We replaced “mineral elements” with “macronutrients”. The correction has been made in the manuscript.
Abstract:
23 – In what context did you use the word "healthy"? Please explain. Generally, such terms are not used, you omit them later in Materials and Methods
Response: Our intention is to show that the physical condition of the geese is consistent. The correction has been made in the manuscript.
24-25 – Please indicate in any way about the combinations of Na and Cl in these mentioned diets. It gives the impression that there were 6 diets when there were actually 9, please consider re-description to clarify this point.
Response: We have re-written this part in line 24-25.
27 – No comma between "yield" and "and". Check out the entire manuscript in this regard.
Response: The correction has been made.
28 – This sentence "affected intestinal morphology" - it's too general. The effect was only for crypt depth in jejunum, please verify it.
Response: We have re-written this part in line 27-28.
29-30 – Please verify, other levels of Na and Cl did not differ from those mentioned, e.g. 0.10 and 0.15
Response: We are sorry for the lack of a clear explanation. According to the results, the jejunal crypt depth was not affected by a single factor (Na or Cl), but by the interaction of sodium and chloride (Na×Cl). And 0.20%Na×0.25%Cl had the smallest jejunal crypt depth.
31-32 – similarly to 0.15 and 0.25, they did not differ from those mentioned
Response: There was no statistically significant difference between the 0.20%Na×0.25%Cl group and the 0.15%Na×0.25%Cl group, but the 0.20%Na×0.25%Cl group had the smallest tibial strength in value.
33-34 – G1, G2, G8 did not differ significantly from G4
Response: As mentioned above, this was the interaction of two factors, so what we can find is that Ca content of bone in 0.15%Na×0.25%Cl group is the highest among the nine groups.
36 – "intestine" too general
Response: The correction has been made.
36-37 – please rephrase "Therefore, we do not advocate excessive supplementation of Na or Cl in the diet, especially too many."
Response: We have re-written this part in line 37.
39 – “be” to “should be” - consider a change
Response: The correction has been made.
Introduction:
49 – Please change "body" to "organism" throughout the manuscript
Response: The correction has been made.
59-60 – no reference to “In previous studies, dietary changes in Na and Cl had not influenced the body weight 59 of 29-70-day-old geese.”
Response: We added the reference in line 61.
63- Please specify "and the first effects may be leg and foot deformities." It seems to be about FPD and this problem in geese is not that important, or please specify it referring to the country of maintenance etc.. Differences can be really significant between regions, countries, this is important.
Response: At present, there are few studies on goose FPD, we list the following two. These two studies confirmed that the environment can affect the incidence of FPD in geese, especially in environments without a swimming pool.
Liao, S. C., Lu, P. X., Shen, S. Y., Hsiao, C. C., Lien, C. Y., Wang, S. D. & Tu, P. A. (2021). Effects of different swimming pool conditions and floor types on growth performance and footpad dermatitis in indoor-reared white roman geese. Animals, 11(6), 1705.
Boz, M. A., Sarıca, M. & Yamak, U. S. (2017). Effect of production system on foot pad dermatitis (FPD) and plumage quality of geese. European Poultry Science, 81.
63-65 – In general, consider whether this passage is relevant, especially since the work is about geese, not broilers as above in general. You need to make it more specific, as per the previous comment.
Response: We are sorry for the lack of a clear explanation. We have added relevant explanations in line 65-67.
71-73 “After three to four weeks of consuming a low sodium diet, the micro-villi surface area more than doubled as a result of an increase in the number and surface area of epithelial cells.” Relatively vague sentence, please rephrase
Response: We have re-written this part in line 74-76.
74 – expected effects, hypothesis?? This is a really important point!
Response: Thank you for your constructive and helpful suggestion. We added this part in line 79-82.
Materials and Methods:
There are some important information gaps, in particular in section 2.1 (the subtitle should include diets). What were the geese?? Breed, crossbreed, where did they come from, how were they fed until day 29?? It is necessary to add information on how the birds were kept. Was a lighting program used? Where were they kept? Maintenance is extremely important! Did the birds have access to the enclosure? What was the bird density?
Response: We are sorry for the lack of important information. We have supplemented this part in 2.1.
Also, the description of the diets is a bit confusing. Perhaps it is worth presenting in what configurations the diets were used (G1-G9). Which diets were identical in Na content? eg G1, G2, G3 had 0.10 etc... and G1, G1, G4, G5 identical in Cl content. Just to make it clear to the reader. Was there a control diet?
Response: Water and feed were provided ad libitum during the experiment. We have re-written this part in 2.1.
89-90 and 92 there is information in the text on chemical analyzes of diets, but it concerns Na, K, Cl. In table 1 there is information on "nutrients levels", for example, Crude protein, Crude fiber ect. But nowhere in the work does it show how these levels were determined. Analytical, calculations? There are also no results regarding the K analysis, and such were done.
Response: We are sorry for the lack of important information. We have supplemented this in line 103-108 and table 1.
111 – six geese from each treatment, i.e. in total, the average was from how many birds? In the table it is n=6 but this indicates that there were a total of 6 birds from each treatment one of the replications or how 6x6 so n=36? Please elaborate on this in each paragraph.
Response: We picked one goose from each pen, that is, each repetition. There were 9 treatments in total, each with 6 replicates, for a total of 54 geese. We have supplemented the information in line 118,125,131.
116 – how was Wall Thickness measured???
Response: The muscle layer and serosa layer were involved, and wall thickness was measured from the intestine's outside to their junction.
127 – where are the results regarding VH:CD ratio??
Response: We have supplemented this part of the data in Table 4.
129 – from how many birds were bones taken, n=???
Response: One goose per replicate, 54 geese in total. We have supplemented the information in line 143.
147 - how were the outliers removed, what were/was the criteria used?
Response: There were few outliers in the experiment, for individual outliers, we choose to keep and participate in the analysis.
Results:
156 – remove the subtitle "Growth performance" or add other titles as appropriate to the paragraphs
Response: The correction has been made.
160-162 – there was a trend for Cl, please take this into account instead of using the word increased, which is assumed to imply a significant difference
Response: As Reviewer suggested that, we have made corrections
162-163 please add where the interaction has not been noted.
Response: We have re-written this part according to the Reviewer’s suggestion.
167 – the tendency. Please mark it
Response: We have re-written the sentence.
169 – indicate which groups were not significantly different from 0.20 Na and 0.25 Cl
Response: We have supplemented the information in line 187-189.
175- Please change "tibial strength" to breaking strength of tibia
Response: The correction has been made.
176 – if there is a Na x Cl interaction, please report these results, not the Na and Cl main results
Response: We have supplemented the information in line 200-201.
179-180 – I suggest, add in parentheses what group it was, it will be easier to locate it in the table. If so, apply to the entire manuscript.
Response: Thank you for your constructive and helpful suggestion. We added this part in manuscript.
181-182 – note that as the Na level increased, the calcium initially increased and then decreased. Verify this sentence again.
Response: The correction has been made in line 207-208.
182-183 – verify this sentence, because according to table 5 there is a tendency (tibbia thickness for Na and P % for Cl)
Response: We have re-written this part in line 196-197 and 209-210.
213 – explanation of the abbreviations Ca and P seems unnecessary
Response: We have made a correction and removed this section.
Discussion:
217- it is worth adding a comparison to the level of 0.40
Response: We have supplemented the information in line 239-240.
221-222 What a “previous observations”??
Response: We have re-written the sentence in line 245-247.
224 – What a “diffrent electorlyte??”
Response: We have re-written the sentence in line 249-250.
231 – it would be advantageous to provide information on feed intake between the groups. Feed consumption is an important parameter in this study and should be considered seriously.
Response: We have supplemented this section in line 259-260.
233-234 – this sentence is unclear and needs re-verification “However, there was no difference among the levels in the NaCl supplemented group (0.127 % -0.509 %)”
Response: We have re-written the sentence in 260-264.
249-250 - indicate that the result was significant only for jejunum
Response: We have supplemented the information in line 275-277.
250-254 – an important parameter is the VH:CD ratio, I suggest adding it and discussing it as this parameter best characterizes the development of the intestines. You should definitely consider that.
Response: Thank you for your valuable and thoughtful comments. We added VH:CD ratio results in the manuscript. We did not find significant differences from the analysis results. We added the corresponding discussion in the line 277-281.
274 – but there was a tendency for Na and Cl
Response: We have re-written this part in 307-313.
Conclusions:
287- verify this for Ca concentration. G9 and, for example, G2 did not differ significantly from each other, and the level of Na was lower.
Response: We have rechecked and re-written the sentence.
287-289 – too bold statement when only differences in crypt depth dudeanum and no ratio VH:CD were observed
Response:Thank you for your valuable and thoughtful comments. We have re-written this part in conclusions.
290-291 – here you contradicted the previous sentence
Response: We have re-written the sentence.
291 – I would change "to sum up" to "in conclusion".
Response: The correction has been made.

Reviewer 2 Report
Dear Authors,
your manuscript is very interesting from a health and zootechnical economic point of view. It is very well prepared. Everything is clear and understandable.
Nowadays is the ongoing selection of birds in the direction of achieving their maximum egg or meat efficiency, led to a weakening of the immune function of their organisms and worse handling of stress factors. Therefore, the use-value of gees is affected not only by their origin, but also by the environmental and social conditions in which they reside, which are related to the rearing system. An increase in nutritional and zoo-hygienic requirements is associated with this.
One of the most prevalent health problems in intensive poultry farming is tibial malformations and anomalies, which pose a serious hazard to the growth and development of poultry. It can cause difficulty in poultry movement, affect feed intake, reduce meat quality, and bring significant economic losses to the poultry industry. Sodium and chloride are essential electrolytes in the body, mainly found in extracellular fluid, and play a critical role in maintaining osmotic pressure and acid-base balance of bodily fluids. However, excessive addition of sodium and chloride may bring many adverse effects, such as increasing water consumption and land salinization, and even threatening poultry health. From this point of view, the presented monograph is very current and can contribute to the increase of knowledge of poultry husbandry and feeding management. This research explored the effects of sodium and chloride levels in diet on slaughter performance, 14 intestinal development and tibia quality of 29-70 days old geese.
The presented manuscript has 13 pages. It is written clearly and comprehensibly. The authors, in the Introduction chapter, were erudite, concentrating on the information on electrolytes, its meaning the body’s system of animals. The objectives of the manuscript are formulated clearly. A main part of the work consists of results and discussions. The results of the manuscript are presented in only one point and using tables. I suggest to divide results in individual points, e.g. Carcass Measurements, Digestive Tract Index, Intestinal Morphology, Bone Strength: Breaking Bones, Bone Mineral Content. In the discussion chapter, the authors compare their own results with the results of other authors and leads a professional controversy. I also suggest to divide on points as well as The Results. There are 38 literary sources. But I suggest to cite newer literature. The conclusions are clear and clearly formulated.
Conclusion:
The manuscript and its presented results are the result of creative activity of the authors, it has all the attributes of scientific work of this character in terms of its own presentation of the publication activity of the authors accepted in the international community.
Based on a thorough study of the manuscript, I recommend, that this document be accepted after minor revision.
Author Response
Response to Reviewer 2 Comments
Thank you for your encouragement of the research. On behalf of my co-authors, we appreciate you very much for your positive and constructive comments and suggestions on our manuscript entitled “Effects of Dietary Sodium and Chloride on Slaughter Performance, Digestive Tract Development and Tibia Mineralization of Geese” (ID: animals-2155543). Those comments are all valuable and very helpful for revising and improving our paper, as well as the important guiding significance to our researches. We have studied comments carefully and have made correction. We have added corresponding subheadings to the Results and Discussion. Revised portion are marked in yellow highlighting in the paper.

Round 2
Reviewer 1 Report
Dear Authors,
thank you for considering my comments. I am satisfied and hope that I have been helpful in correcting your manuscript.